# LEARNING THE UNLEARNABLE: ADVERSARIAL AUGMENTATIONS SUPPRESS UNLEARNABLE EXAMPLE ATTACKS

## ABSTRACT

Unlearnable example attacks are data poisoning techniques that can be used to safeguard public data against unauthorized training of deep learning models. These methods add stealthy perturbations to the original image, thereby making it difficult for deep learning models to learn from these training data effectively. Current research suggests that adversarial training can, to a certain degree, mitigate the impact of unlearnable example attacks, while common data augmentation methods are not effective against such poisons. Adversarial training, however, demands considerable computational resources and can result in non-trivial accuracy loss. In this paper, we introduce the *UEraser* method, which outperforms current defenses against different types of state-of-the-art unlearnable example attacks through a combination of effective data augmentation policies and loss-maximizing adversarial augmentations. In stark contrast to the current SOTA adversarial training methods, *UEraser* uses adversarial augmentations, which extends beyond the confines of $\ell_p$ perturbation budget assumed by current unlearning attacks and defenses. It also helps to improve the model's generalization ability, thus protecting against accuracy loss. *UEraser* wipes out the unlearning effect with loss-maximizing data augmentations, thus restoring trained model accuracies. Interestingly, *UEraser-Lite*, a fast variant without adversarial augmentations, is also highly effective in preserving clean accuracies. On challenging unlearnable CIFAR-10, CIFAR-100, SVHN, and ImageNet-subset datasets produced with various attacks, it achieves results that are comparable to those obtained during clean training. We also demonstrate the efficacy of *UEraser* against possible adaptive attacks. Our code is open source and available to the deep learning community [1].

## 1 INTRODUCTION

Deep learning has achieved great success in fields such as computer vision [14] and natural language processing [8], and the development of various fields now relies on large-scale datasets. While these datasets have undoubtedly contributed significantly to the progress of deep learning, the collection of unauthorized private data for training these models now presents an emerging concern. Recently, numerous poisoning methods [11, 16, 28, 31, 35] have been proposed to add imperceptible perturbations to images. These perturbations [16, 35] have the potential to impede the learning process of models, preventing them from accurately learning the original data features in order to preserve privacy. It is commonly perceived that the only effective defense against unlearnable examples are adversarial training algorithms [16, 31, 11]. Popular data augmentation methods such as CutOut [9], MixUp [39], and AutoAugment [5], however, have all been demonstrated to be ineffective defenses.

Current methods of unlearnable attacks involves the specification of an $\ell_p$ perturbation budget, where $p \in \{2, \infty\}$ in general. Essentially, they constrain the added perturbation to a small $\epsilon$-ball of $\ell_p$-distance from the source image, in order to ensure stealthiness of these attacks. Adversarial training defenses [21, 11] represent a defense mechanism that seeks to counteract the bounded perturbations from such unlearnable attacks. However, large defensive perturbations comes with

---

[1]Link to follow

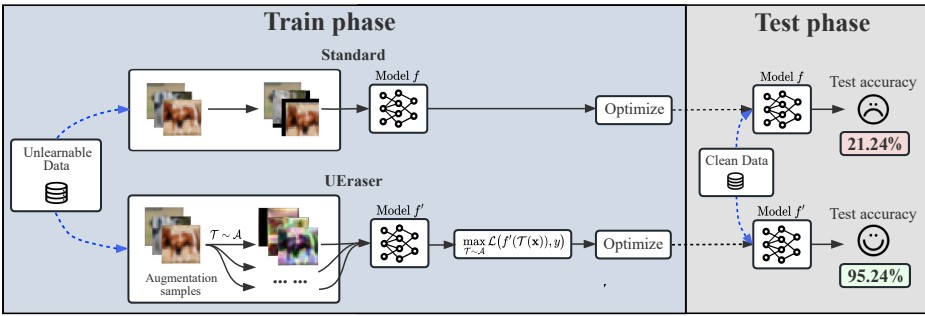

Figure 1: A high-level overview of *UEraser* for countering unlearning poisons. Note that *UEraser* recovers the clean accuracy of unlearnable examples by data augmentations. The reported results are for EM [16] unlearnable CIFAR-10 with an $\ell_\infty$ perturbation budget of $8/255$.

significant accuracy degradations. This prompts the inquiry of the existence of effective defense mechanisms that leverage threat models that are outside the purview of attackers. Specifically, *can we devise effective adversarial policies for training models that extend beyond the confines of the $\ell_p$ perturbation budgets?*

In this paper, we thus propose *UEraser*, which performs loss-maximizing data augmentation, to defense against unlearning poisons. *UEraser* challenges the preconception that data augmentation is not an effective defense against unlearning poisons. *UEraser* expands the perturbation distance far beyond traditional adversarial training, as data augmentation policies do not confine themselves to the $\ell_p$ perturbation constraints. It can therefore effectively disrupt "unlearning shortcuts" formed by attacks within narrow $\ell_p$ constraints. Yet, the augmentations employed by *UEraser* are natural and realistic transformations extensively utilized by existing works to improve the models' ability to generalize. This, in turn, helps in avoiding accuracy loss due to perturbations used by adversarial training that could potentially be out-of-distribution. Finally, traditional adversarial training is not effective in mitigating unlearning poisons produced by adaptive attacks [11], while *UEraser* is highly resiliant against adaptive attacks with significantly lower accuracy reduction.

In summary, our work has three main contributions:

- It extends adversarial training beyond the confines of the $\ell_p$ perturbation budgets commonly imposed by attackers into data augmentation policies.

- We propose *UEraser*, which introduces an effective adversarial augmentation to wipe out unlearning perturbations. It defends against the unlearnable attacks by maximizing the error of the augmented samples.

- *UEraser* is highly effective in wiping out the unlearning effect on eight state-of-the-art (SOTA) unlearning attacks, outperforming existing SOTA defense methods.

- We explore the adaptive attacks on *UEraser* and explored additional combinations of augmentation policies. It lays a fresh foundation for future competitions among unlearnable example attack and defense strategies.

Unlearnable example attacks bear great significance, not just from the standpoint of privacy preservation, but also as a form of data poisoning attack. It is thus of great significance to highlight the shortcomings of current attack methods. Perhaps most surprisingly, even a well-known unlearnable attack such as EM [16] is unable to impede the effectiveness of *UEraser*. By training a ResNet-18 model from scratch using exclusively CIFAR-10 unlearnable data produced with EM (with an $\ell_\infty$ budget of $8/255$), *UEraser* achieves exceptional accuracy of $95.24\%$ on the clean test set, which closely matches the accuracy achievable by standard training on a clean training set. This suggests that existing unlearning perturbations are tragically inadequate in making data unlearnable, even with adaptive attacks that employs *UEraser*. By understanding their weaknesses, we can anticipate how malicious actors may attempt to exploit them, and prepare stronger safeguards against such threats. We hope *UEraser* can help facilitate the advancement of research in these attacks and defenses.

## 2 RELATED WORK

**Adversarial examples and adversarial training.** Adversarial examples deceive machine learning models by adding adversarial perturbations, often imperceptible to human, to source images, leading to incorrect classification results [13, 30]. White-box adversarial attacks [30] maximize the loss of a source image with gradient descent on the defending model to add adversarial perturbations onto an image to maximize its loss on the model. Effective methods to gain adversarial robustness usually involve adversarial training [21], which leverages adversarial examples to train models. Adversarial training algorithms thus solve the min-max problem of minimizing the loss function for most adversarial examples within a perturbation budget, typically bounded in $\ell_p$. Recent years have thus observed an arms race between adversarial attack strategies and defense mechanisms [3, 4, 36, 37].

**Data poisoning.** Data poisoning attacks manipulate the training of a deep learning model by injecting malicious and poisoned examples into its training set [1, 29]. Data poisoning methods [2, 24] achieve their malicious objectives by stealthily replacing a portion of training data, and successful attacks can be triggered with specially-crafted prescribed inputs. Effective data poisoning attacks typically perform well on clean data and fail on data that contains triggers [19].

**Unlearnable examples.** Unlearnable examples attacks are a type of data poisoning methods with bounded perturbation that aims to make learning from such examples difficult. Unlike traditional data poisoning methods, unlearnable examples methods usually require adding imperceptible perturbations to all examples [11, 16, 28, 31, 35]. NTGA [38] simulates the training dynamics of a generalized deep neural network using a Gaussian process and leverages this surrogate to find better local optima with improved transferability. Error-minimizing (EM) [16] poison generates imperceptible perturbations with a min-min objective, which minimizes the errors of training examples on a trained model, making them difficult to learn by deep learning models. By introducing noise that minimizes the error of all training examples, the model instead learns "shortcut" of such perturbations, resulting in inability to learn from such data. Unlike EM, Adversarial poisoning (TAP) [10] considers the adversarial sample generation and uses the error maximization process to generate adversarial samples for the purpose of unlearnability. Hypocritical perturbations (HYPO) [31] follows a similar idea but uses a pretrained surrogate rather than the above min-min optimization. As the above method cannot defend against adversarial training, Robust Error-Minimizing (REM) [11] uses an adversarially-trained model as an adaptively attack to generate stronger unlearnable examples. INF [33] enables samples from different classes to share non-discriminatory features to improve resistance to adversarial training. Linear-separable Synthetic Perturbations (LSP) [35] reveals that if the perturbations of unlearnable samples are assigned to the corresponding target label, they are linearly separable. It thus proposes linearly separable perturbations in response to this characteristic and show great effectiveness. Autoregressive poisoning (AR) [28] proposes a generic perturbation that can be applied to different datasets and architectures. The perturbations of AR are generated from dataset-independent processes. One pixel shortcut (OPS) [34] is a targeted availability poisoning attack that perturbs only one pixel of an image, generating an effective availability poisoning attack against traditional adversarial training methods.

**Data augmentations.** Data augmentation techniques increase the diversity of training data by applying random transformations [18] (such as rotation, flipping, cropping, *etc.*) to images, thereby improving the model's generalization ability. Currently, automatic search-based augmentation techniques such as TrivialAugment [22] and AutoAugment [5], can further improve the performance of trained DNNs by using a diverse set of augmentation policies. TorMentor [25], an image-augmentation framework, proposes fractal-based data augmentation to improve model generalization. Current unlearnable example methods [11, 16, 28, 31, 35] demonstrate strong results under an extensive range of data augmentation methods. Despite prevailing beliefs on their ineffectiveness against unlearnable examples, *UEraser* challenges this preconception, as it searches for adversarial policies with loss-maximizing augmentations and achieves the state-of-art defense performance against existing unlearnable example attacks.

## 3 THE *UEraser* DEFENSE

### 3.1 PRELIMINARIES ON UNLEARNABLE EXAMPLE ATTACKS AND DEFENSES

**Attacker.** We assume the attacker has access to the original data they want to make unlearnable, but cannot alter the training process [19]. Typically, the attacker attempts to make the data unlearnable

by adding perturbations to the images to prevent trainers from using them to learn a classifier that generalize well to the original data distribution. Formally, suppose we have a dataset consisting of original clean examples $\mathcal{D}_{\text{clean}} = \{(\boldsymbol{x}_1, y_1), \ldots, (\boldsymbol{x}_n, y_n)\}$ drawn from a distribution $\mathcal{S}$, where $\boldsymbol{x}_i \in \mathcal{X}$ is an input image and $y_i \in \mathcal{Y}$ is its label. The attacker thus aims to construct a set of sample-specific unlearning perturbations $\boldsymbol{\delta} = \{\boldsymbol{\delta_x} | \boldsymbol{x} \in \mathcal{X}\}$, in order to make the model $f_{\boldsymbol{\theta}} : \mathcal{X} \to \mathcal{Y}$ trained on the *unlearnable examples* set $\mathcal{D}_{\text{ue}}(\boldsymbol{\delta}) = \{(\boldsymbol{x} + \boldsymbol{\delta_x}, y) \mid (\boldsymbol{x}, y) \in \mathcal{D}_{\text{clean}}\}$ perform poorly on a test set $\mathcal{D}_{\text{test}}$ sampled from $\mathcal{S}$:

$$\max_{\boldsymbol{\delta}} \mathbb{E}_{(\boldsymbol{x}_i, y_i) \sim \mathcal{D}_{\text{test}}}[\mathcal{L}(f_{\boldsymbol{\theta}^\star(\boldsymbol{\delta})}(\boldsymbol{x}_i), y_i)], \text{ s.t. } \boldsymbol{\theta}^\star(\boldsymbol{\delta}) = \operatorname*{argmin}_{\boldsymbol{\theta}} \sum_{(\boldsymbol{x}_i, y_i) \in \mathcal{D}_{\text{ue}}(\boldsymbol{\delta})} \mathcal{L}(f_{\boldsymbol{\theta}}(\hat{\boldsymbol{x}}_i), y_i), \quad (1)$$

where $\mathcal{L}$ is the loss function, typically the softmax cross-entropy loss. For each image, the noise $\boldsymbol{\delta}_i$ is bounded by $\|\boldsymbol{\delta}_i\|_p \leq \epsilon$, where $\epsilon$ is a small perturbation budget such that it may not affect the intended utility of the image, and $\|\cdot\|_p$ denotes the $\ell_p$ norm. Table 1 provides samples generated by unlearnable example attacks and their corresponding perturbations (amplified with normalization).

Table 1: The visualization of unlearned examples and perturbations of eight poisoning methods on CIFAR-10.

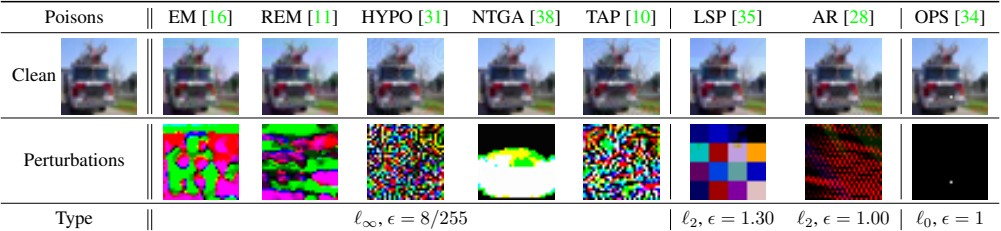

| Poisons | EM [16] | REM [11] | HYPO [31] | NTGA [38] | TAP [10] | LSP [35] | AR [28] | OPS [34] |
|---|---|---|---|---|---|---|---|---|
| Clean | | | | | | | | |
| Perturbations | | | | | | | | |
| Type | | | $\ell_\infty, \epsilon = 8/255$ | | | $\ell_2, \epsilon = 1.30$ | $\ell_2, \epsilon = 1.00$ | $\ell_0, \epsilon = 1$ |

**Defender.** The goal of the defender is to ensure that the trained model learns from the poisoned training data, allowing the model to be generalized to the original clean data distribution $\mathcal{D}_{\text{clean}}$. The attacker assumes full control of its training process. In our context, we thus assume that the attacker's policy is to perform poison removal on the image, in order to ensure the trained model generalizes even when trained on poisoned data $\mathcal{D}_{\text{ue}}$. It has been shown in [16, 31, 11] that Adversarial training [21] is effective against unlearnable examples, which optimizes the following objective:

$$\arg\min_{\boldsymbol{\theta}} \mathbb{E}_{(\hat{\boldsymbol{x}}, y) \sim \mathcal{D}_{\text{ue}}} \left[ \max_{\|\boldsymbol{\delta}_{\text{adv}}\|_p \leq \epsilon} \mathcal{L}(f_{\boldsymbol{\theta}}(\hat{\boldsymbol{x}} + \boldsymbol{\delta}_{\text{adv}}), y) \right]. \quad (2)$$

Specifically for each image $\hat{\boldsymbol{x}} \in \mathcal{D}_{\text{ue}}$, it finds an adversarial perturbation $\boldsymbol{\delta}_{\text{adv}}$ that maximizes the classifier loss. It then performs gradient descent on the maximal loss to optimize for the model parameters $\boldsymbol{\theta}$. A model trained on the unlearnable set $\mathcal{D}_{\text{ue}}$ in this manner thus demonstrates robustness to perturbations in the input, and can generalize to clean images.

## 3.2 ADVERSARIAL AUGMENTATIONS

Geirhos *et al.* [12] reveal that models tend to learn "shortcuts", *i.e.*, unintended features in the training images. These shortcuts negatively impact the model's generalization ability. Intuitively, unlearnable examples make the model inclined to ignore original features in images by injecting strongly linearly separable perturbations into the data, creating unlearning shortcuts to impede learning from the poisoned examples [35]. Subsequently, existing adversarial training defenses [16, 31, 11] attempt to remove these shortcuts from training images with adversarial perturbations. This is done to counter the effects of the unlearning perturbations.

It is natural to think that augmentation policies may be a dead end against unlearnable attacks, as none of the existing strong data augmentation methods show significant effectiveness (Table 2). Adversarial training can also be viewed as a practical data augmentation policy, which presents an interesting perspective as it allows the model to choose its own policy in the form of $\ell_p$-bounded perturbations adaptively. However, it poses a considerable challenge due to its use of large defensive perturbations, often resulting in reduced accuracy. This begs the question of whether new defense mechanisms can leverage *unseen* threat models that unlearnable attacks may be unable to account for.

Inspired by this, we introduce *UEraser*, which performs adversarial augmentations polices that preserves to the semantic information of the images rather than adding $\ell_p$-bounded adversarial noise.

Our objective is a bi-level optimization, where the inner level samples image transformation policies $\mathcal{T}(\cdot)$ from a set of all possible augmentations $\mathcal{A}$, in order to maximize the loss, and the outer level performs model training with adversarial polices:

$$\arg\min_{\boldsymbol{\theta}} \mathbb{E}_{(\boldsymbol{x},y)\sim\mathcal{D}_{\text{ue}}} \left[ \max_{\mathcal{T}\sim\mathcal{A}} \mathcal{L}(f_{\boldsymbol{\theta}}(\mathcal{T}(\boldsymbol{x})), y) \right]. \tag{3}$$

Intuitively, *UEraser* finds the most "adversarial" augmentation policies for the current images, and use that to train the model in order to prevent unlearnable "shortcuts" from emerging during model training. Compared to adversarial training methods that confine the defensive perturbations within a small $\epsilon$-ball of $\ell_p$ distance, here we adopt a different approach that allows for a more aggressive distortion. Moreover, augmentation policies also effectively preserve the original semantics in the image. By maximizing the adversarial loss in this manner, the model can thus avoid training from the unlearning "shortcuts" and instead learn from the original features.

To generate augmentation policies with high-level of distortions, we select PlasmaTransform [25], and TrivialAugment [22], two modern suites of data augmentation policies, and ChannelShuffle in sequence, to form a strong pipeline of data augmentations polices. PlasmaTransform performs image distortion with fractal-based transformations. TrivialAugment provide a suite of natural augmentations which shows great generalization abilities that can train models with SOTA accuracies. Finally, ChannelShuffle swaps the color channels randomly, this is added to further increase the aggressiveness of adversarial augmentation policies.

Interestingly, using this pipeline without the error-maximization augmentation sampling can also significantly reduce the effect of unlearning perturbations. We denote this method as *UEraser-Lite*, as it requires only 1 augmentation sample per training image. Compared to *UEraser*, although *UEraser-Lite* may not perform as well as *UEraser* on most datasets, it is more practical than both *UEraser* and adversarial training due to its faster training speed.

Finally, we provide an algorithmic overview of *UEraser* in Algorithm 1. It accepts as input the poisoned training dataset $\mathcal{D}_{\text{ue}}$, batch size $B$, randomly initialized model $f_{\boldsymbol{\theta}}$, number of training epochs $N$, number of loss-maximizing augmentation epochs $W$, learning rate schedule $\boldsymbol{\alpha} = [\boldsymbol{\alpha}_1, \boldsymbol{\alpha}_2, \ldots, \boldsymbol{\alpha}_N]$, number of repeated sampling $K$, and a suite of augmentation policies $\mathcal{A}$. For each sampled mini-batch $\boldsymbol{x}, \boldsymbol{y}$ of data points from the dataset, it applies $K$ different random augmentation policies for each image in $\boldsymbol{x}$, and compute the corresponding loss values for all augmented images. It then selects for each image in $\boldsymbol{x}$, the maximum loss produced by its $K$ augmented variants. The algorithm then uses the averaged loss across the same mini-batch to perform gradient descent on the model parameters. Finally, the algorithm returns the trained model parameters $\boldsymbol{\theta}$ after completing the training process.

---

**Algorithm 1** Training with *UEraser*.

---

1: **function** UERASER($\mathcal{D}_{\text{ue}}, B, f_{\boldsymbol{\theta}}, N, W, \boldsymbol{\alpha}, K, \mathcal{A}$)
2:     **for** $n \in [1, \ldots, N]$ **do**
3:         **if** $n \geq W$ **then** $K \leftarrow 1$ **end if**     $\triangleright$ Disable adversarial augmentations after warmup.
4:         **for** $(\boldsymbol{x}, \boldsymbol{y}) \sim \text{minibatch}(\mathcal{D}_{\text{ue}}, B)$ **do**         $\triangleright$ Mini-batch sampling.
5:             **for** $i \in [1, \ldots, B]$ **do**         $\triangleright$ For each image in mini-batch
6:                 **for** $j \in [1, \ldots, K]$ **do**         $\triangleright$ Repeat $K$ augmentations.
7:                     $\text{aug} \sim \mathcal{A}$         $\triangleright$ Sample augmentation policy
8:                     $\mathbf{L}_{ij} \leftarrow \mathcal{L}(f_{\boldsymbol{\theta}}(\text{aug}(\boldsymbol{x}_i)), \boldsymbol{y}_i)$     $\triangleright$ Evaluate the loss for the augmented image.
9:                 **end for**
10:                 $\mathbf{L}_i^{\text{adv}} \leftarrow \max_{j\in[1,\ldots,K]} \mathbf{L}_{ij}$     $\triangleright$ Find the augmented image with maximum loss.
11:             **end for**
12:             $\boldsymbol{\theta} \leftarrow \boldsymbol{\theta} - \boldsymbol{\alpha}_n \nabla_{\boldsymbol{\theta}} \frac{1}{B} \sum_{i\in[1,\ldots,B]} \mathbf{L}_i^{\text{adv}}$     $\triangleright$ SGD on mini-batch of max-loss images.
13:         **end for**
14:     **end for**
15:     **return** $\boldsymbol{\theta}$
16: **end function**

---

## 4 EXPERIMENTAL SETUP & RESULTS

**Datasets.** We select four popular datasets for the evaluation of *UEraser*, namely, CIFAR-10 [17], CIFAR-100 [17], SVHN [23], and an ImageNet [7] subset. Following the setup in EM [16], we use the first 100 classes of the full dataset as the ImageNet-subset, and resize all images to $224 \times 224$. We evaluate the effectiveness of *UEraser* by examining the accuracies of the trained models on clean test examples, *i.e.*, the higher the clean test accuracy, the greater its effectiveness. By default, all target and surrogate models use ResNet-18 [14] if not otherwise specified. We explore the effect of *UEraser* on partial poisoning (Section 4), larger perturbation budgets (Section 4), different network architectures (Section 4), transfer learning (Appendix C), and perform ablation analyses in Section 4. Finally, Appendix B provides additional sensitivity analyses.

**Training settings.** We train the CIFAR-10, CIFAR-100 and ImageNet-subset model for 300 epochs, and SVHN model for 150 epochs, as unlearnable datasets can notably slow convergence. We adopt standard random cropping and flipping for all experiments by default as standard training baselines and introduce additional augmentations as required by the compared methods. For the optimizer, we use SGD with a momentum of 0.9 and a weight decay of $5 \times 10^{-4}$. By default, we use a cosine annealing learning rate schedule with an initial learning rate of 0.01. We follow the dataset setup in respective attacks, where all unlearning perturbations are bounded within $\ell_\infty = 8/255$ or $\ell_2 = 1.00(1.30)$. For *UEraser*, we divided the training process into two parts for speed: the adversarial augmentation process, and the standard training process. In the first stage, we used the loss-maximizing augmentations for training, with a default number of repeated samples $K = 5$ (as the input to Algorithm 1). In the second stage, we used the *UEraser-Lite* process which sets $K = 1$. This approach allows us to keep a balance between suppressing the emergence of unlearning shortcuts and training speed. We further explore full training with loss-maximizing augmentation in Section 4, which attains the highest known test accuracies. Finally, Table 2 shows the effect of *UEraser* on eight different unlearnable methods. Additional information regarding the training setup and hyperparameters can be found in Appendix A. The details of the attack and defense baselines are available in Appendix G.

**Main Evaluation.** To demonstrate the effectiveness of *UEraser*, we select 8 SOTA sample-wise unlearnable example attacks: Error-Minimization (EM) [16], Hypocritical Perturbations (HYPO) [31], NTGA [38], Robust Error-Minimization (REM) [11], Adversarial poisons (TAP) [10], Linear-separable Synthetic Perturbations (LSP) [35], Autoregressive Poisoning (AR) [28], and One-pixel shortcut (OPS) [34]. Experimental results show that *UEraser* has achieved better results than the SOTA defense methods: Image shortcut squeezing (ISS) [20] and adversarial training [21]. For EM, REM, NTGA, HYPO and TAP, We use the same model (ResNet-18) as the surrogate and target model. All unlearnable methods have a poisoning rate of $100\%$. From the experimental results of *UEraser* on CIFAR-10 dataset shown in Table 2, we can conclude that *UEraser* can achieve better defensive results than ISS and adversarial training in most cases. Notably, with *UEraser-Max*, when the loss-maximizing augmentation is used throughout the training phase, the model achieves SOTA accuracy rates under most unlearnable example attacks.

In Table 3, We further validate the effect of *UEraser* on CIFAR-100, SVHN and ImageNet-subset. We select the popular method (EM) and the latest attacks for the experiments (LSP and AR). Note that since *UEraser* increases the diversity of the data with strong augmentations, it requires more training epochs to achieve converged accuracies. All results are thus evaluated after 300 training epochs for CIFAR-100 and ImageNet-subset and 150 training epochs for SVHN. Experimental results also show that *UEraser* achieves SOTA results on all three datasets.

| Original | PlasmaTransform | ChannelShuffle | TrivialAugment | Ueraser |
|---|---|---|---|---|

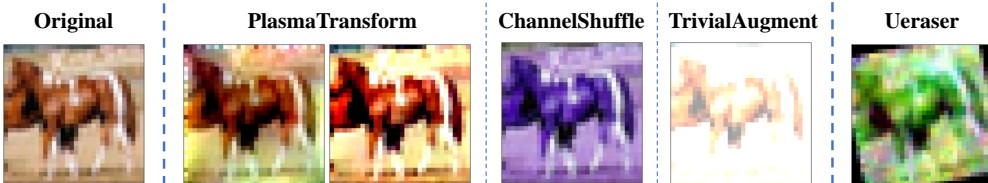

Figure 2: The Visualization of *UEraser* augmentations.

**Partial Poisoning.** In practical scenarios, attackers may only have partial control over the training data [16], thus it is more practical to consider the scenario where only a part of the data is poisoned. We adopt EM [16] and LSP [35] on the CIFAR-10 dataset as an example for our discussions.

Table 2: Clean test accuracies (%) of ResNet-18 models trained on CIFAR-10 unlearnable examples with various attack and defenses. "ST" denotes standard training with random crop and flip augmentations. When training on clean data with *UEraser-Lite* and *UEraser* the accuracy is 93.94% and 93.66% respectively. Note that ISS [20] contains three strategies (Grayscale, JPEG, and BDR), and we report the results of their best strategy. More specifically, Grayscale for EM and REM, JPEG for HYPO, LSP, and AR. "AT" means standard $\ell_\infty$ adversarial training with $\epsilon = 8/255$.

| Methods | ST | CutOut [9] | MixUp [39] | CutMix [9] | *UEraser-Lite* | *UEraser* | *UEraser-Max* | ISS [20] | AT |
|---|---|---|---|---|---|---|---|---|---|
| EM | 21.21 | 19.30 | 58.51 | 22.40 | 90.78 | 93.38 | 95.24 | 92.27 | 83.02 |
| REM | 25.44 | 26.54 | 29.02 | 34.48 | 85.49 | 91.02 | **92.54** | 91.34 | 82.87 |
| HYPO | 70.38 | 69.04 | 74.25 | 67.12 | 85.67 | 87.59 | **88.67** | 84.77 | 85.49 |
| NTGA | 18.15 | 13.78 | 20.59 | 12.91 | 78.29 | 84.41 | **87.94** | 72.65 | 70.05 |
| TAP | 6.27 | 9.88 | 15.46 | 14.21 | 83.29 | **84.17** | 82.47 | 83.05 | 81.19 |
| LSP | 14.95 | 10.67 | 41.52 | 23.84 | 84.92 | 85.07 | **94.95** | 82.71 | 84.27 |
| AR | 11.75 | 11.90 | 11.40 | 11.23 | 87.12 | 88.64 | **89.82** | 84.67 | 84.16 |
| OPS | 14.69 | 52.98 | 49.27 | 64.72 | 68.50 | 73.22 | **81.84** | 77.81 | 11.08 |

Table 3: Clean test accuracies (%) of *UEraser* on CIFAR-100, SVHN, and ImageNet-subset. The results of ISS [20] are from the best strategy (Grayscale for EM and JPEG for LSP). '†' denotes the ImageNet-subset of 100 classes. When training on clean data with UEraser-Lite the accuracy is 70.84%, 93.06%, and 67.39% for CIFAR-100, SVHN, and ImageNet-subset respectively.

| Dataset | Clean | Methods | Standard | *UEraser-Lite* | *UEraser* | *UEraser-Max* | ISS [20] | AT |
|---|---|---|---|---|---|---|---|---|
| CIFAR-100 | 74.83 | EM | 13.15 | 70.04 | 71.14 | 72.20 | 54.92 | 41.36 |
| | | REM | 3.61 | 65.84 | 67.17 | **68.51** | 55.60 | 40.03 |
| | | LSP | 4.09 | 67.38 | 68.42 | **68.64** | 51.35 | 39.11 |
| SVHN | 96.12 | EM | 10.58 | 88.73 | 91.22 | 91.08 | 89.91 | 86.05 |
| | | REM | 11.08 | 87.79 | **90.72** | 88.36 | 90.58 | 85.37 |
| | | LSP | 14.56 | 92.64 | **92.79** | 90.64 | 84.06 | 86.71 |
| ImageNet† | 72.44 | EM | 3.08 | 34.83 | **45.67** | 44.07 | 32.46 | 41.85 |
| | | REM | 5.63 | 35.06 | 41.05 | **42.14** | 40.98 | 40.36 |
| | | LSP | 38.52 | 59.63 | 62.78 | **63.50** | 40.25 | 46.80 |

Following the same setup, we split varying percentages from the clean data to carry out unlearnable poisoning and mix it with the rest of the clean training data for the target model training. *UEraser* is applied during model training to explore its effectiveness against partial poisoning. Figures 3a and 3b, show that when the poisoning ratio is low ($< 40\%$), the effect of the poisoning is negligible. Another type of partial dataset attack scenario is the selection of a targeted class to poison. We thus poison all training samples of the $9^{th}$ label ("truck"), and Figures 3c and 3d shows the prediction confusion matrices of ResNet-18 trained on CIFAR-10. In summary, *UEraser* demonstrates significant efficacy in partial poisoning scenarios.

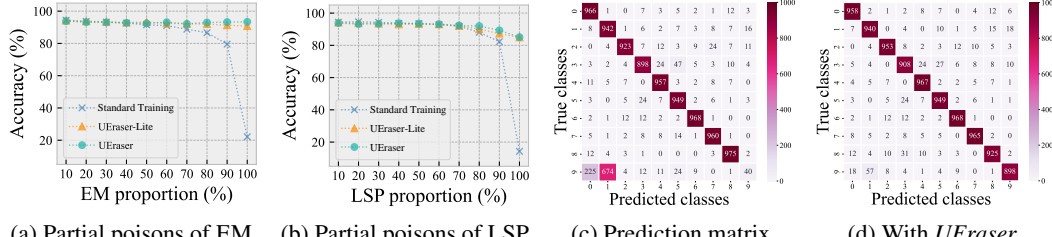

(a) Partial poisons of EM.  (b) Partial poisons of LSP.  (c) Prediction matrix.  (d) With *UEraser*.

Figure 3: The defensive efficacy of *UEraser* against partial poisoning. (a) EM with different poisoning ratios; (b) LSP with different poisoning ratios. (c), (d) Prediction confusion matrices on the clean test set of ResNet-18 trained on CIFAR-10 with an unlearnable class (*the $9^{th}$ label 'truck'*). (c) Standard training; (d) *UEraser* training.

**Adaptive Poisoning.** Since *UEraser* is composed of multiple data augmentations, we should consider possible adaptive unlearnable example attacks which may leverage *UEraser* to craft poisons against it. We therefore evaluate *UEraser* in a worst-case scenario where the adversary is fully aware of our defense mechanism, in order to reliably assess the resilience of *UEraser* against potential adaptive attacks. Specifically, we design an adaptive unlearning poisoning attack by introducing an additional

data augmentation during the training, We adopt the error-minimization (EM) attack [16] as an example. The EM unlearning objective solves the following min-min optimization:

$$\arg \min_{\boldsymbol{\delta}} \min_{\boldsymbol{\theta}} \mathbb{E}_{(\boldsymbol{x},y)\sim \mathcal{D}_{\text{clean}}}[\mathcal{L}(f_{\boldsymbol{\theta}}(\boldsymbol{x}+\boldsymbol{\delta}_{\boldsymbol{x}}),y)], \quad (4)$$

where $\|\boldsymbol{\delta}\|_p \le \epsilon$. Similar to the REM [11] that generates adaptive unlearnable examples under adversarial training, each image $\boldsymbol{x}$ optimizes its unlearning perturbation $\boldsymbol{\delta}_{\boldsymbol{x}}$ before performing adversarial augmentations:

$$\arg \min_{\boldsymbol{\delta}} \min_{\boldsymbol{\theta}} \mathbb{E}_{(\boldsymbol{x},y)\sim \mathcal{D}_{\text{clean}}}[\mathcal{L}(f_{\boldsymbol{\theta}}(\mathcal{T}_{\text{adv}}(\boldsymbol{x}+\boldsymbol{\delta}_{\boldsymbol{x}})),y)], \quad (5)$$

where $\mathcal{T}_{\text{adv}}(\cdot)$ denotes the adversarial augmentations with *UEraser*. We select *UEraser-Lite* and *UEraser-Max* as the objective of adaptive poisoning for our experiments, and the results are shown in Table 4. The hyperparameters of the augmentations employed in all experiments are kept consistent with those of *UEraser*. We observe that the adaptive augmentation of unlearning poisons do not significantly reduce the effectiveness of *UEraser*. As it encompasses a diverse set of augmentation policies and augmentation intensities, along with loss-maximizing augmentation sampling, adaptive poisons are hardly effective. Moreover, we speculate that the affine and cropping transformations in TrivialAugment can cause unlearning perturbations to be confined to a portion of the images, which also limits the effectiveness of unlearning poisons. Because of the aggressiveness of the augmentation in image transformations extend beyond the $\ell_p$ bounds, adaptive poisons do not perform as well under *UEraser* as they do against REM. To summarize, it is challenging for the attacker to achieve successful poisoning against *UEraser* even if it observes the possible transformations taken by the augmentations.

Table 4: Adaptive poisoning with EM on CIFAR-10.

| Methods | Standard | *UEraser-Lite* | *UEraser* | *UEraser-Max* |
|---|---|---|---|---|
| Baseline | 21.21 | 90.78 | 93.38 | 95.24 |
| + *UEraser-Lite* | 29.36 | 81.24 | 87.68 | 89.55 |
| + *UEraser-Max* | 35.24 | 60.15 | 71.04 | 80.28 |

**Larger Perturbation Scales.** Will the performance of *UEraser* affected by large unlearnable perturbations? To verify, we evaluate the performance of *UEraser* on unlearnable CIFAR-10 dataset with even larger perturbations. We use the example of error-maximizing (EM) attack and increase the $\ell_\infty$ perturbation from $8/255$ to $24/255$ to examine the efficacy of *UEraser* on a more severe unlearning perturbation scenario. We also include adversarial training (AT) as a defense baseline with a perturbation bound of $8/255$. The experimental results in Table 5 confirm the effectiveness of *UEraser* under large unlearning noises.

Table 5: Increasing the perturbation budget $\epsilon$ of the EM and LSP attack on unlearnable defenses.

| Type | Budget | Standard | *UEraser-Lite* | *UEraser* | *UEraser-Max* | AT |
|---|---|---|---|---|---|---|
| EM ($\ell_\infty$) | 8/255 | 21.24 | 90.78 | 93.38 | **95.24** | 86.24 |
| | 16/255 | 22.63 | 86.65 | 89.24 | **90.71** | 83.12 |
| | 24/255 | 21.05 | 82.40 | 84.59 | **86.07** | 79.31 |
| LSP ($\ell_2$) | 1.30 | 14.95 | 84.92 | 85.07 | **94.95** | 84.27 |
| | 1.74 | 15.83 | 78.65 | 81.26 | **87.51** | 77.38 |
| | 2.17 | 14.27 | 47.30 | 65.19 | **74.62** | 70.24 |

**Resilience against Architecture Choices.** Can *UEraser* show resilience against architecture choices? In a realistic scenario, we need to train the data with different network architectures. We thus explore whether *UEraser* can wipe out the unlearning effect under different architectures. Table 6 shows the corresponding results. It is clear that *UEraser* is capable of effectively wiping out unlearning poisons, across various network architectures.

**Augmentation Options.** In this part, we investigate the impact of the *UEraser* augmentation policies (visualized in Figure 2) on the mitigation of unlearning effects. We conduct experiments with the unlearnable examples from CIFAR-10 generated by the EM [16] method, and Table 7 explore effectiveness of the various combinations of augmentation policies. ISS [20] discovered that for the unlearnable examples generated by the EM attack, a grayscale filter easily removes the unlearning poisons. Additionally, setting the value of each channel to the average of all color channels or to

Table 6: Clean test accuracies (%) of different architectures (ResNet-50 [14], DenseNet-121 [15], MobileNet-V2 [27]). Note that EM is tested with $\ell_\infty = 8/255$ and LSP is tested with $\ell_2 = 1.30$.

| | Methods | ResNet-50 | DenseNet-121 | MobileNet-V2 |
|---|---|---|---|---|
| | Clean | 94.39 | 95.14 | 94.20 |
| EM | Standard | 25.17 | 34.91 | 31.75 |
| | *UEraser-Lite* | 89.56 | 91.20 | 90.55 |
| | *UEraser* | 89.74 | 92.37 | 89.92 |
| LSP | Standard | 14.94 | 22.71 | 20.04 |
| | *UEraser-Lite* | 84.17 | 86.22 | 83.24 |
| | *UEraser* | 85.56 | 87.19 | 84.85 |

the value of any color channel also considerably achieves the same effect. However, we show that using only ChannelShuffle does not yield satisfactory results. We have also discovered an interesting phenomenon: PlasmaTransform and ChannelShuffle are essential for mostly restoring the accuracies, whereas TrivialAugment, can be substituted with a similar policy, *i.e.* AutoAugment [5]. Only when the three policies are employed together can the effect of unlearning poisons be effectively wiped out. This also proves that the *UEraser* policies are effective and reasonable. Recall that we also found the adoption of loss-maximizing augmentation results in an overall improvement on all unlearning poisons. Hence, the utilization of loss-maximizing augmentation during training serves as an effective means to mitigate the challenges of training with unlearning examples and improve the model's clean accuracy.

**Adversarial Augmentations and Loss-Maximizing Epochs.** From Algorithm 1, the training of *UEraser* is affected by two hyperparameters, namely, the numbers of repeated augmentation samples $K$ per image and the epochs of loss-maximizing augmentation $W$. For CIFAR-10, The clean accuracy of the unlearnable examples can be improved to around $80\%$ after 50 epochs of training using loss-maximizing augmentation. Regarding the number of samples $K$ (by default $K = 5$), increasing it further enhances the suppression of unlearning shortcuts during model training, but also more likely to lead to gradient explosions at the beginning of model training. Therefore, it may be necessary to apply learning rate warmup or gradient clipping with increased number of repeated sampling. Larger $K$ can also results in higher computational costs, as it result in more samples per image for training. We provide a sensitivity analysis of the number of repeated sampling $K$ in Appendix B.

Table 7: Ablation analysis *UEraser* on clean and EM unlearnable CIFAR-10. Note that all hyperparameters are the same, except for the *UEraser* augmentation policy which varies. 'P', 'C', 'T', and 'A' denote "PlasmaBrightness", "ChannelShuffle", "TrivialAugment", and "AutoAugment" [5] respectively. 'AA' denotes the adversarial augmentation used in *UEraser-Max*.

| Policies | Clean | Unlearnable |
|---|---|---|
| Standard Training | 94.78 | 21.24 |
| + P | 94.47 | 29.48 |
| + T | 94.47 | 48.81 |
| + P + C | 94.15 | 62.17 |
| + P + T | 95.22 | 48.05 |
| + C + T | 94.40 | 69.24 |
| + C + P + A | 94.04 | 85.60 |
| + C + P + T | 93.94 | 90.78 |
| + C + P + T + AA | 93.66 | 93.38 |

## 5 CONCLUSION

Using the intuition of disrupting the unlearning perturbation with perturbations beyond the $\ell_p$ budgets, we propose a simple yet effective defense method called *UEraser*, which can mitigate unlearning poisons and restore clean accuracies. *UEraser* achieves robust defenses on unlearning poisons with simple data augmentations and adversarial augmentation policies. Similar to adversarial training, it employs loss-maximizing augmentation to further eliminate the impact of unlearning poisons. Our comprehensive experiments on eight state-of-the-art unlearnable example attacks demonstrate that *UEraser* outperforms existing countermeasures such as adversarial training [16, 11, 31]. We also evaluate adaptive poisons and transfer learning on *UEraser*. Our results suggest that existing unlearning perturbations are tragically inadequate in making data unlearnable. By understanding the weaknesses of existing attacks, we can anticipate how malicious actors may attempt to exploit them, and prepare stronger safeguards against such threats. We hope *UEraser* can help facilitate the advancement of research in these attacks and defenses.

## 6 REPRODUCIBILITY STATEMENT

We provide an open-source implementation of all *UEraser* variants in the supplementary material. All experiments in the paper uses public datasets, *e.g.*, CIFAR-10, CIFAR-100, SVHN, and ImageNet-subset. Following the README file, users can run *UEraser* experiments on their own device to reproduce the results shown in paper with the model architectures and hyperparameters in Appendix A.

## 7 ETHICS STATEMENT

Unlearnable examples can potentially serve as a protective mechanism for private data, but malicious exploitation of such examples may lead to data poisoning. Our study focuses on thwarting data poisoning attacks that limit the generalization ability of a model through malicious poisoning.

Despite acknowledging that a malicious party may exploit the approach suggested in this paper, we believe that the ethical approach for the open-source deep learning community is not to withhold information but rather to increase awareness of these risks. Moreover, our proposed defenses presents potential means of circumventing such attacks, thereby inspiring more effective privacy protection techniques for defenders.

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

# A  EXPERIMENTAL SETUP

## A.1  DATASETS

**CIFAR-10** consists of 60,000 $32 \times 32$ resolution images, of which 50,000 images are the training set and 10,000 are the test set. This dataset contains 10 classes, each with 6000 images.

**CIFAR-100** is similar to CIFAR-10. It has 100 classes, Each class has 600 images of size $32 \times 32$, of which 500 are used as the training set and 100 as the test set.

**SVHN**, derived from Google Street View door numbers, is a dataset of cropped images containing sets of Arabic numerals '0-9'. The dataset consists of 73,257 digit images in the training set and 26,032 digit images in the test set.

**ImageNet-subset** refers to a dataset constructed with the 100 classes from ImageNet, following the setup in [16] for fair comparisons. The training set comprises 20,000 images, and the test set contains 4,000 images.

Table 8 shows the detail specifications of these datasets.

Table 8: Overview of the specifications of datasets used in this paper.

| Dataset | Input size | Train-set | Test-set | Classes |
|---|---|---|---|---|
| CIFAR-10 | $32 \times 32 \times 3$ | 50,000 | 10,000 | 10 |
| CIFAR-100 | $32 \times 32 \times 3$ | 50,000 | 10,000 | 100 |
| SVHN | $32 \times 32 \times 3$ | 73,257 | 26,032 | 10 |
| ImageNet-subset | $224 \times 224 \times 3$ | 20,000 | 4,000 | 100 |

## A.2  MODELS AND HYPERPARAMETERS

We evaluate *UEraser* using a standard ResNet-18 [14] architecture by default, and extend experiments to standard ResNet-50, DenseNet-121, and MobileNet-v2 in Table 6. In all the experiments, we used a stochastic gradient descent (SGD) optimizer with a momentum of 0.9. Table 9 provide the default hyperparameters used to evaluate *UEraser* and  on unlearnable examples.

Table 9: Default hyperparameters for all attacks.

| Hyperparameters | CIFAR-10 | CIFAR-100 | SVHN | ImageNet-subset |
|---|---|---|---|---|
| Default hyperparameters for all attacks | | | | |
| Initial learning rate | 0.01 | 0.01 | 0.01 | 0.01 |
| Learning rate schedule | Cosine | Cosine | Cosine | Cosine |
| Weight decay | 5e-4 | 5e-4 | 5e-4 | 5e-4 |
| Epochs $E$ | 300 | 300 | 150 | 300 |
| Batch size $B$ | 128 | 128 | 128 | 128 |
| Additional default hyperparameters for *UEraser* | | | | |
| Repeated sampling $K$ | 5 | 5 | 5 | 5 |
| Loss-maximizing epochs $W$ | 50 | 30 | 30 | 30 |
| Additional default hyperparameters for *UEraser-Max* | | | | |
| Repeated sampling $K$ | 5 | 5 | 5 | 5 |
| Loss-maximizing epochs $W$ | 300 | 300 | 150 | 300 |

## A.3  STANDARD AUGMENTATION

For CIFAR-10, SVHN, and CIFAR-100 baselines to compare against, we perform data augmentation via random flipping, and random cropping to $32 \times 32$ images on each image. For the ImageNet-subset,

we perform data augmentation with a 0.875 center cropping, followed with a resize to $32 \times 32$, and random flipping for each image.

## A.4 UNLEARNING PERTURBUTION BUDGETS

Following the attack literatures, we use the default perturbation budgets defined in the respective implementations. The attacks, EM [16], REM [11], NTGA [38], TAP [10] and HYPO [31], all have a permitted perturbation bound of $\ell_\infty = 8/255$ for each image. Additionally, the LSP [35] and AR [28] attacks permit $\ell_2 = 1.00/\ell_2 = 1.30$ and OPS [34] attacks permit $\ell_0 = 1$.

## A.5 ADVERSARIAL TRAINING

For comparison, the baseline defenses against the eight methods (EM, REM, NTGA, TAP, HYPO, LSP, and AR) on CIFAR-10 employ PGD adversarial training [21], following the evaluation of [11]. The adversarial training perturbation bounds used were $\ell_\infty = 8/255$ for all experiments.

## A.6 HYPERPARAMETERS FOR *UEraser*

*UEraser* comprises three composite augmentations (PlasmaTransform, ChannelShuffle, TrivialAugment). We implement augmentations using Kornia[2]. Here, TrivialAugment uses the default hyperparameters, Table 10 shows the hyperparameter settings for the remaining augmentations. Finally, Figure 4 provides the visualization of the augmentation effects on different dataset examples.

Table 10: Default hyperparameters for *UEraser-Lite* / *UEraser* / *UEraser-Max* augmentations.

| Augmentations | PlasmaBrightness | PlasmaContrast | ChannelShuffle |
|---|---|---|---|
| Probability of use $p$ | 0.5 | 0.5 | 0.5 |
| Roughness | (0.3/0.1/0.1, 0.7) | (0.3/0.1/0.1, 0.7) | - |
| Intensity | (0.5/0.0/0.0, 1.0) | - | - |
| Same on batch | False | False | False |

Table 11: Default hyperparameters for *UEraser* augmentations on ImageNet-subset.

| Augmentations | RandomGrayscale | PlasmaBrightness | PlasmaContrast | ChannelShuffle |
|---|---|---|---|---|
| Probability of use $p$ | 0.5 | 0.5 | 0.5 | 0.5 |
| Roughness | - | (0.1, 0.7) | (0.1, 0.7) | - |
| Intensity | - | (0.0, 1.0) | - | - |
| Same on batch | False | False | False | False |

## B SENSITIVITY ANALYSIS

Here, we provide a sensitivity analysis of the number of repeated sampling $K$ using in adversarial augmentation. We also explore the effect of increasing the number of training epochs under *UEraser*. Taking the example of unlearnable CIFAR-10 produced with EM [16], Table 12 shows the results of *UEraser* with various combinations of different numbers of repeated augmentation sampling $K$ and the total number of training epochs $E$. Higher $K$ values can effectively improve the defense performance of *UEraser*, with a training cost increasing proportionally with $K$. More training epochs can also improve the performance of *UEraser*, and even matches the test accuracy of training with clean data. Finally, Figure 5 provides the train and test accuracy curves *w.r.t.* the number of training epochs for different $(E, K)$ configurations.

---

[2]Documentation: https://kornia.readthedocs.io/en/latest/augmentation.module.html.

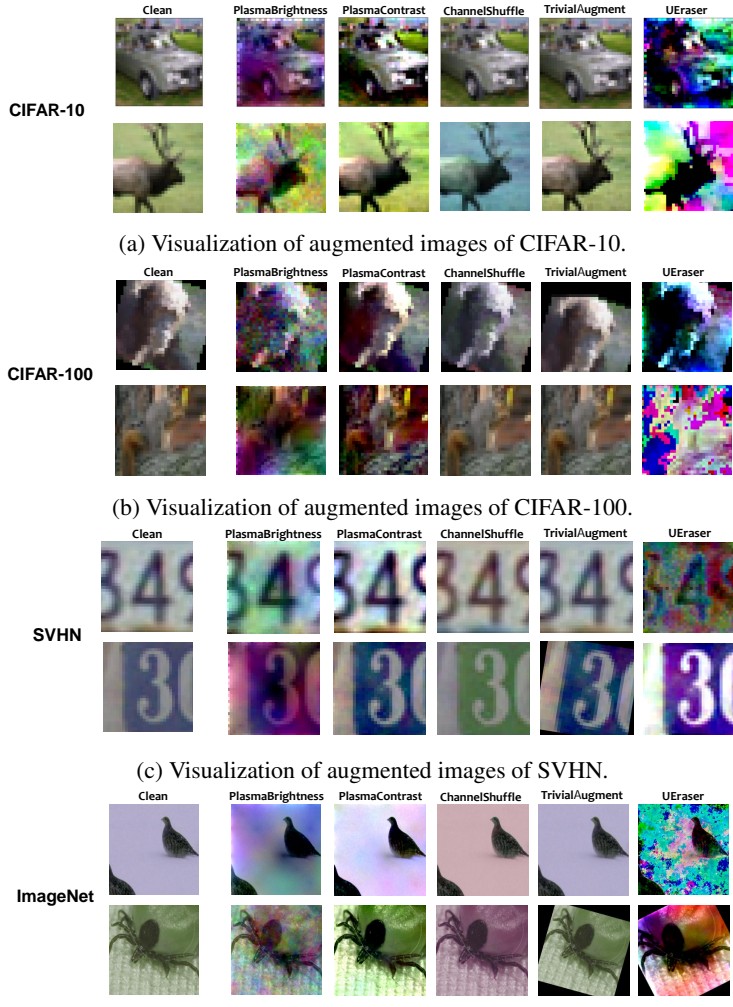

(a) Visualization of augmented images of CIFAR-10.

(b) Visualization of augmented images of CIFAR-100.

(c) Visualization of augmented images of SVHN.

(d) Visualization of augmented images of ImageNet.

Figure 4: The visualization of various augmentations on different datasets.

## C  TRANSFER LEARNING

In this section, we aim to explore the impact of transfer learning [6, 32] on the efficacy of unlearnable examples. We hypothesize that pretrained models may learn certain in-distribution features of the unaltered target distribution, it may be able to gain accuracy even if the training set contains unlearning poisons.

To this end, we adopt a simple transfer learning setup, where we use the pretrained ImageNet ResNet-18 model available in the torchvision repository [26]. To fit the expected input shape of the feature extractor, we upsampled the input images to $224 \times 224$. The final fully-connected classification layer of the pretrained model was replaced with a randomly initialized one with 10 logit outputs. We then fine-tune the model with unlearnable CIFAR-10 training data using either *UEraser-Lite* (Figure 6d) or *UEraser* (Figure 6e). We also further explored fine-tuning on unlearnable data with our defenses. For control references, we fine-tuned a model with either clean training data (Figure 6b) or the unlearnable data (Figure 6c), and also trained a randomly initialized model from scratch with poisoned training data (Figure 6a).

First, Figure 6a highlights that simply upsampling the unlearnable samples to use more compute and have larger feature maps does not significantly weaken the unlearning attack, with test accuracy increased to $34\%$ after upsampling. Figure 6c shows that fine-tuning with unlearnable examples

Table 12: Clean test accuracies (%) of different numbers of repeated augmentation sampling $K$ for *UEraser*. Note that $K = 1$ denotes *UEraser-Lite*. For *UEraser*, the number of error-maximizing epochs $W = 50$, whereas $W = E$ for *UEraser-Max*. "—" means $K = 1$ is not applicable for *UEraser-Max*. The unlearnable training data is generated with EM on CIFAR-10, and a standard ResNet-18 trained on this data attains a test accuracy of $21.21\%$.

| $K$ | *UEraser* ($W = 50$) | | *UEraser-Max* ($W = E$) | |
|---|---|---|---|---|
| | $E = 200$ | $E = 300$ | $E = 200$ | $E = 300$ |
| 1 | 90.78 | 94.19 | — | — |
| 2 | 92.76 | 94.79 | 91.32 | 95.25 |
| 3 | 92.91 | 95.01 | 91.40 | **95.85** |
| 4 | 93.02 | **95.14** | 91.26 | 95.59 |
| 5 | 93.38 | 94.83 | 92.12 | 95.24 |
| 6 | 93.24 | 94.78 | 93.17 | 95.36 |
| 7 | 93.16 | 94.50 | 93.58 | 94.79 |
| 8 | 93.23 | 94.87 | 93.95 | 94.84 |
| 9 | 92.86 | 94.58 | 93.92 | 94.66 |
| 10 | 93.03 | 94.83 | 94.06 | 94.57 |

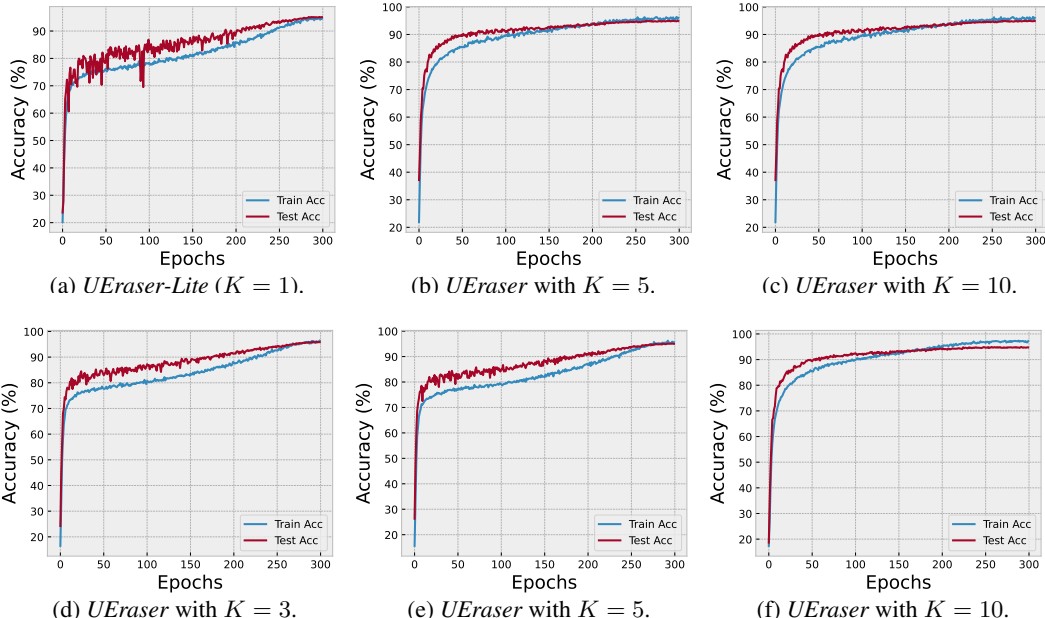

(a) *UEraser-Lite* ($K = 1$).    (b) *UEraser* with $K = 5$.    (c) *UEraser* with $K = 10$.

(d) *UEraser* with $K = 3$.    (e) *UEraser* with $K = 5$.    (f) *UEraser* with $K = 10$.

Figure 5: Train and test accuracy curves *w.r.t.* the number of training epochs. The subfigures correspond to *UEraser* under $K \in \{1, 5, 10\}$ and *UEraser-Max* under $K \in \{3, 5, 10\}$. Recall that $K$ is the number of repeated augmentation sampling.

can improve the clean test accuracy from $22\%$ to $66\%$. Most importantly, *UEraser* successfully eliminates the negative impact of unlearning poisons, which enables the model to utilize pretrained knowledge effectively. This enables the fine-tuned model to achieve a test accuracy of approximately $95\%$ as shown in Figure 6e.

## D   EXTEND EVALUATIONS ON CIFAR-100

We additionally evaluate the effect of *UEraser* on the CIFAR-100 in larger perturbation scenarios. As shown in Table 13, *UEraser* still has favorable defense results under larger perturbation scenarios. This further illustrates the effectiveness of *UEraser*.

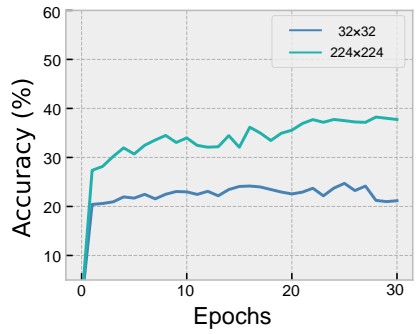

(a) Test accuracies by training from scratch.

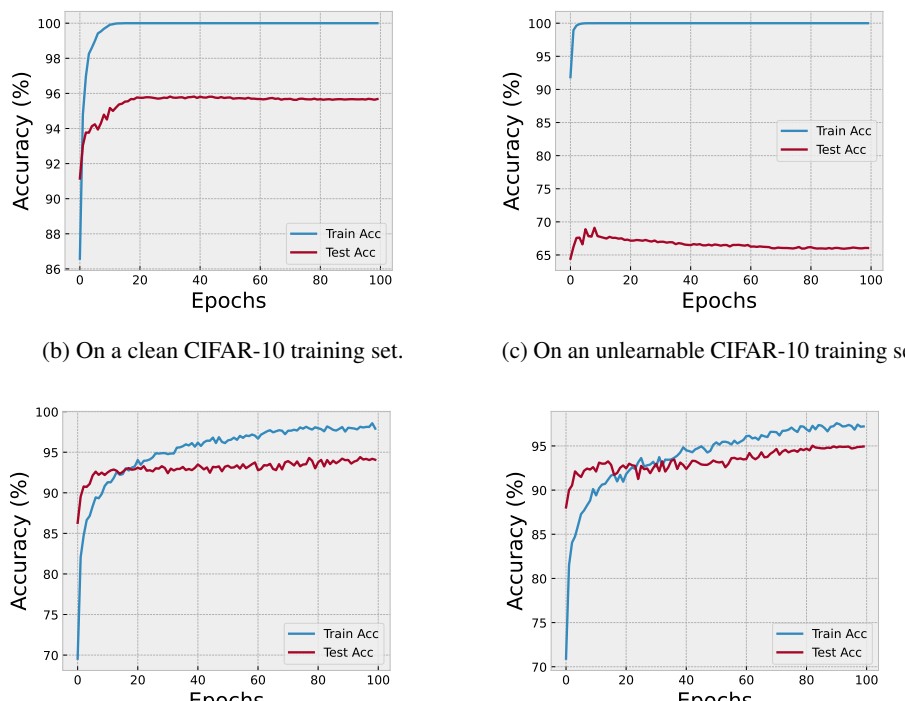

(b) On a clean CIFAR-10 training set.  (c) On an unlearnable CIFAR-10 training set.

(d) On an unlearnable CIFAR-10 training set under *UEraser-Lite*.  (e) On an unlearnable CIFAR-10 training set under *UEraser*.

Figure 6: Accuracies *w.r.t.* the number of fine-tuning epochs for a pretrained ImageNet ResNet-18 on different upsampled CIFAR-10 datasets. (b) Fine-tuning the pretrained model on a clean training set. (c) Fine-tuning the pretrained model with unlearnable training set generated with EM [16]. (d) Fine-tuning on the unlearnable data with *UEraser-Lite*. (e) Fine-tuning on the unlearnable data with *UEraser*. (a) Test accuracies by training from scratch on the unlearnable data with different input dimensions.

# E    ADAPTIVE POISONING FOR ISS

Compared to *UEraser*, which applies an adversarial strategy, the defense effectiveness of ISS is more affected in the scenario of adaptive poisoning. In the adaptive scenario Table 14, the best accuracy of ISS recovery is 69.48%, which is significantly lower than UEraser-max (80.28%).

Table 13: Increasing the perturbation budget $\epsilon$ of the EM and LSP attack on CIFAR-100.

| Type | Budget | Standard | *U-Lite* | *U* | *U-Max* | AT |
|------|--------|----------|----------|-----|---------|-----|
| EM ($\ell_\infty$) | 8/255 | 13.15 | 70.04 | 71.14 | **72.20** | 61.08 |
| | 16/255 | 12.07 | 57.02 | 62.57 | **66.68** | 52.63 |
| | 24/255 | 12.28 | 50.27 | 52.54 | **54.81** | 49.91 |
| LSP ($\ell_2$) | 1.30 | 4.09 | 67.38 | 68.42 | **68.64** | 57.72 |
| | 1.74 | 3.61 | 57.08 | 65.35 | **67.94** | 52.83 |
| | 2.17 | 2.94 | 43.79 | 62.83 | **64.37** | 50.27 |

Table 14: Clean test accuracy of ISS under adaptive poisoning on EM.

| Datesets | Standard | Grayscale | JPEG | G&J |
|----------|----------|-----------|------|-----|
| Baseline | 21.05 | 93.01 | 81.05 | 83.06 |
| + Grayscale | 17.81 | 16.60 | 76.71 | 74.16 |
| + JPEG | 17.11 | 89.18 | 83.11 | 82.85 |
| + G&J | 48.93 | 46.29 | 69.48 | 66.26 |

## F  COMPUTATIONAL OVERHEAD

Table 15: Time costs per epoch for *UEraser* training. Note that all experimental time overheads were calculated on a Tesla V100.

| Datesets | Normal | Grayscale | JPEG | *UEraser-Lite* | *UEraser-Max* ($K = 5$) |
|----------|--------|-----------|------|----------------|-------------------------|
| CIFAR-10 | 21s | 24s | 23s | 32s | 2min 32s |
| CIFAR-100 | 29s | 33s | 34s | 40s | 2min 07s |
| SHVN | 23s | 23s | 24s | 34s | 2min 31s |
| ImageNet-subset | 3min 41s | 3min 43s | 3min 57s | 4min 49s | 9min 37s |

## G  ATTACK AND DEFENSE BASELINES

We use five baseline attacks and two exisiting SOTA defenses for evaluation and comparisons in our experiments (Table 16). Each attack method is implemented from their respective official source code for a fair comparison. We adopt experimental setup identical to the original publications, and use perturbation budgets described in Appendix A.4. For defenses, we compare *UEraser* variants against the current SOTA techniques, image shortcut squeezing [20] and adversarial training [21]. The compared defenses (ISS and adversarial training) respectively follow the original source code and PGD adversarial training [21].

Table 16: Attack and defense methods and respective links to open source repositories.

| Name | Open Source Repository |
|---|---|
| Attacks | |
| Error-minimizing attack (EM) [16] | https://github.com/HanxunH/Unlearnable-Examples/ |
| Robust error-minimizing attack (REM) [11] | https://github.com/fshp971/robust-unlearnable-examples |
| Hypocritical perturbations (HYPO) [31] | https://github.com/TLMichael/Delusive-Adversary |
| Neural tangent generalization attacks (NTGA) [38] | https://github.com/lionelmessi6410/ntga |
| Adversarial examples make strong poisons (TAP) [10] | https://github.com/lhfowl/adversarial_poisons/tree/main |
| Linear-separable synthetic perturbations (LSP) [35] | https://github.com/dayu11/Availability-Attacks-Create-Shortcuts |
| Autoregressive poisoning (AR) [28] | https://github.com/psandovalsegura/autoregressive-poisoning |
| One-Pixel Shortcut (OPS) [34] | https://github.com/cychomatica/One-Pixel-Shotcut |
| Defenses | |
| Image shortcut squeezing [20] | https://github.com/liuzrcc/ImageShortcutSqueezing |
| Adversarial Training [21] | https://github.com/lafeat/apbench/blob/main/madrys.py |

