# OpenReview forum: "Learning the Unlearnable: Adversarial Augmentations Suppress Unlearnable Example Attacks"
_ICLR.cc/2024/Conference — ICLR 2024 Conference Withdrawn Submission_

### Official Review · Reviewer_3r2o · 2023-10-29

**Soundness:** 2 fair
**Presentation:** 3 good
**Contribution:** 2 fair
**Rating:** 3
**Confidence:** 4

**Summary:**

Unlearnable example attacks are used to safeguard public data against unauthorized training of deep learning models. To break the data protection of unlearnable example methods, this paper introduces UEraser, an adversarial augmentation (PlasmaTransform, TrivialAugment and ChannelShuffle) based method to help model to learn semantic information from unlearnable example. Similarly to adversarial training, UEraser tries to find an augmentation to maximize the training loss rather than to find a perturbation to maximize the training loss (adversarial training does).

**Strengths:**

1. This paper focuses on the under-explored topic to learn semantic information from unlearnable examples, which is interesting and impressive.
2. This paper is organized logically and written clearly. The visualization results of different unlearnable example methods are impressive.
3. Impressive accuracy improvements on on four datasets (CIFAR-10, CIFAR-100, ImageNet, SVHN) compared with standard training.

**Weaknesses:**

1. The motivation and insight of this paper are not clear. As mentioned in Sec. 1 " These perturbations can form “shortcuts” [12, 16] in the training data to prevent training and thus make the data unlearnable in order to preserve privacy.", [12,16] does not reveal that unlearnable perturbations are shortcuts from models. Such two references are not suitable and convincing. Is there any observation or experiment to confirm that unlearnable perturbations are shortcuts from models? The same confusion occurs in Sec. 3.2 " Geirhos et al. [12] reveal that models tend to learn “shortcuts”, i.e., unintended features in the training images.". [12] only expounded that models prefer to learn shortcuts. Whether and why unlearnable perturbations are shortcuts from models is not clear at all.
2. The explanation of why UEraser performs better than adversarial training is self-contradictory. As mentioned in Sec. 3.2, this paper believed that UEraser augmentation policies more effectively preserve the original semantics in the image than adversarial training. However, adversarial training set a $\ell_{p}$ bound to constraint larger change on original input. The distance between adversarial training input and original input is much more small than the distance between UEraser augmented input and original input. In this case, whether UEraser augmentation policies more effectively preserve the original semantics in the image than adversarial training is quite doubtful. This paper should focus on analyzing why UEraser augmentation works and adversarial training does not. More insights should be proposed rather than engineering experiments.
3. As shown in Table 2, CutOut, CutMix does not work on unlearnable examples, but ChannelShuffle does. Not sure if all spatial transformation-based augmentation does not work and color transformation-based augmentation does? If so, why not filter out spatial transformation-based augmentation in TrivialAugment like shear and rotate to improve performance?
4. Clerical errors in Sec. 3.2. "Compared to UEraser, although UEraser-Lite may not perform as well as UEraser on most datasets, it is more practical than both UEraser-Lite and adversarial training due to its faster training speed." should be "Compared to UEraser, although UEraser-Lite may not perform as well as UEraser on most datasets, it is more practical than both UEraser and adversarial training due to its faster training speed."

**Questions:**

1. It is confused about the experimental setup. In Sec. 4 this paper resizes all images of ImageNet-subset to 32x32. However, as mentioned in Table 8, the input size of ImageNet-subset is 224x224x3. Which one is the real experimental setup? In addition, the operation of resize should not be implemented to ImageNet-subset, because the efficient of UEraser should be verified on high-resolution images perturbed by unlearnable methods, which are aligned with the experimental setup of EM and REM.
2. This paper repeats that UEraser-Lite has a fast training speed. How fast it is? I'd like to know the comparison results of execution time between UEraser (UEraser-Lite, UEraser, UEraser-Max) and adversarial training.
3. Why the augmentation of UEraser-Lite is Channleshuffle? Is there any other augmentation (equalize, posterize, plasmabrightness) that can achieve the same result as Channleshuffle?
4. Table 2 and Table 3 should add the results of UEraser trained model on clean data. Considering as an attacker, you have no idea whether the training is clean or not. The results of adversarial training and other unlearnable methods (AR, OPS, TAP, NTGA, HYPO) in Table 3 should be shown out.
5. There is no sensitivity analysis of hyperparameter W. How to pick a suitable value of W and K when deploying UEraser?

---

> ### Author Response · Authors · 2023-11-15
> **Rebuttal by Authors**
>
> Thank you for reviewing our paper and we would like to address your concerns below.
>
> > Confusion regarding "These perturbations can form “shortcuts” [12, 16]"
>
> Thank you for highlighting the confusion
> regarding the term "shortcuts"
> in the context of unlearnable example attacks.
> We agree with the reviewer that [16]
> does not directly state
> that unlearnable perturbations are shortcuts,
> but its optimization of perturbations
> implies such a conclusion,
> as the perturbation added is encouraged
> to form unintended features with loss minimization.
> Instead of [16],
> we could cite [a] with an explicit claim
> on unlearnability shortcuts.
> We also note that [12] is cited
> as it provides the original definition
> of the term "shortcuts".
> We have updated the paper accordingly.
>
> [a]: Yu et al., Availability attacks create shortcuts, SIGKDD 2022.
>
> > The explanation of why UEraser performs better than adversarial training is self-contradictory.
>
> In Section 3.2,
> we intend the sentence
> "... which performs adversarial augmentations polices that preserves to the semantic information of the images rather than adding $\ell_p$-bounded adversarial noise"
> to describe the property of adversarial augmentations
> as preserving semantic information,
> instead of comparing the abilities
> to preserve semantics of both methods.
> We thank the reviewer for pointing this out
> and agree with the reviewer
> that the sentence
> could lead to confusion,
> and have updated the relevant text
> in Section 3.2 accordingly.
>
> While $\ell_p$ bounded perturbations
> do preserve semantics from the perspective human perception,
> from a machine learning perspective,
> when crafted correctly,
> they have a profound impact
> on deep learning algorithms,
> making them unable
> to learn the original features effectively.
> In this sense,
> UEraser policies do preserve semantics better
> than standard training,
> as they can train models
> that learned such semantics
> much more effectively.
>
> > why not filter out spatial transformation-based augmentation ...
>
> The direct use of TrivialAugment is to ensure the plug-and-play and simplicity of UEraser.
> Moreover,
> similar to standard training,
> the use of spatial transformations
> is to help further improve the generalization ability
> of the trained models
> to attain higher clean test accuracies.
>
> > Clerical errors in Sec. 3.2.
>
> Thanks a lot for noticing!
> We have fixed the typos in the updated version.
>
> > Sec. 4 ... resizes all images of ImageNet-subset to 32x32. However, ... Table 8 ... is 224x224x3.
>
> Thank you for pointing this out to us.
> We can confirm that our ImageNet subset results
> are trained on 224x224 images,
> but Section 4 were not updated to reflect this.
>
> > This paper repeats that UEraser-Lite has a fast training speed. How fast it is?
>
> Thank you,
> and we have added the time costs
> of UEraser variants
> versus other defense methods
> in Appendix F.
>
> > Table 2 and Table 3 should add the results of UEraser trained model on clean data.
>
> We kindly note
> that we reported the results of UEraser trained model
> on clean data in the description of Table 2.
> In the updated version of Table 3,
> we have also added the results
> for CIFAR-100, SVHN and ImageNet.
>
> > The results of adversarial training and other unlearnable methods (AR, OPS, TAP, NTGA, HYPO) in Table 3 should be shown out.
>
> Thank you for your constructive suggestions.
> We added the results of adversarial training in Table 3
> in the updated version.
>
> > There is no sensitivity analysis of hyperparameter W. How to pick a suitable value of W and K when deploying UEraser?
>
> A smaller W makes UEraser training faster
> while a larger W gives better results.
> We kindly note that the sensitivity analyses
> of $K \in \{1,2,\ldots,10\}$
> and $W \in \{50, E\}$,
> where $E$ is the number of total epochs,
> are reported in Table 12 of Appendix B.
> UEraser-Max (i.e. $W = E$)
> usually achieves the best results,
> while being the most computationally expensive.
> We chose $K = 5$ by default
> for all experiments that involve
> adversarial augmentations,
> as it offers a good trade-off
> between training speed and accuracy.
> Larger $K$ values
> resulted in lower accuracies,
> as the adversarial augmentations
> may become too aggressive.

---

### Official Review · Reviewer_E5wZ · 2023-10-29

**Soundness:** 3 good
**Presentation:** 3 good
**Contribution:** 3 good
**Rating:** 6
**Confidence:** 2

**Summary:**

This paper considers the following adversarial poisoning problem: An adversary uses training data with "small perturbations (so in particular not hard to distinguish with normal training data" to tamper the training process so that the trained model will generalize poorly in the distribution sense.

The paper proposes a new "data augmentation" mechanism, which is solve objective (3) under various augmentations.

Quite some experiments are performed to demonstrate the effectiveness of the method.

**Strengths:**

Overall this paper is a nice read. There is a clear principle that guides the paper (objective (3)), and the results seem pretty solid. UEraser seems to work pretty well in all experiments.

It is also a bit surprising that "data augmentation" actually works. What is missing in previous work?

**Weaknesses:**

For one thing, the paper seems rather straightforward in principle, we try to solve the objective (3) under various augmentation methods. So as we expand the augmentations, we should get better results.

One thing is what is really the cost (which I don't think the paper has much discussion), do we need a tremendous amount of augmentation in order to make sure the learning does not learn the short cut? Also, what happens if the adversary is aware of the augmentations the training method is using? (so in that sense the paper needs to be more clear about what is the security model -- that is, what knowledge does the adversary has?)

**Questions:**

I don't have specific questions, my main concerns are described above.

---

> ### Author Response · Authors · 2023-11-15
> **Rebuttal by Authors**
>
> Thank you for reviewing our paper and we would like to address your concerns below.
>
> > ... seems rather straightforward in principle, we try to solve the objective (3) under various augmentation methods. So as we expand the augmentations, we should get better results.
>
> We think this can also be taken as a compliment to our work.
> Simple and effective ideas are often the most impactful.
> We hope that our work can inspire future research
> to explore the potential of adversarial augmentation
> in unlearnable example attacks and defenses
>
> > do we need a tremendous amount of augmentation ... ?
>
> Thanks for raising an interesting question.
> While the augmentations we used are not exhaustive,
> we believe they have the potential to be improved further.
> This could be optimized with auto machine learning algorithms,
> and we hope to explore this in the future.
>
> > what happens if the adversary is aware of the augmentations the training method is using ...
>
> We kindly point out that this scenario is considered
> in Section 4 *Adaptive Poisoning* and Table 4.
> It shows that UEraser-Max
> can maintain a good defense
> against adaptive attacks
> using the UEraser variants.

---

### Official Review · Reviewer_6kgo · 2023-10-31

**Soundness:** 3 good
**Presentation:** 3 good
**Contribution:** 3 good
**Rating:** 6
**Confidence:** 3

**Summary:**

This paper tackles unlearnable example attacks in deep learning, introducing "UEraser," a defense method utilizing adversarial data augmentations to neutralize these attacks. UEraser extends the perturbation budget beyond typical attacker assumptions, aiming to maintain model accuracy on poisoned data without significant loss on clean data. A faster variant, "UEraserLite," is also presented. The authors demonstrate UEraser's effectiveness across various state-of-the-art unlearnable example attacks, outperforming existing defenses and showing resilience against adaptive attacks.

**Strengths:**

- The paper introduces "UEraser," a novel defense against unlearnable example attacks using adversarial data augmentations. This approach is novel and extends the perturbation budget beyond typical attacker assumptions, showcasing a new direction in defending against these types of attacks.

- The authors have conducted extensive experiments to validate the effectiveness of UEraser against various state-of-the-art unlearnable example attacks. The results demonstrate that UEraser outperforms existing defense methods and is resilient against potential adaptive attacks, providing a strong empirical basis for the proposed approach.

- The introduction of "UEraserLite" offers a faster and more efficient alternative to UEraser, making the proposed defense more accessible and practical for real-world applications.

**Weaknesses:**

- While the paper presents a novel approach to defending against unlearnable example attacks, the contribution can be considered incremental. The use of adversarial training and data augmentation for defense is not entirely new, and the extension to unlearnable example attacks, while valuable, builds upon existing knowledge and techniques.

- The paper could benefit from exploring additional evaluation of UEraser and UEraserLite, particularly in terms of computational efficiency/time cost compared with existing defenses.

- The paper primarily focuses on empirical results, and a more comprehensive theoretical analysis of why UEraser works and under what conditions it is most effective could strengthen the paper.

**Questions:**

See above.

---

> ### Author Response · Authors · 2023-11-15
> **Rebuttal by Authors**
>
> Thank you for reviewing our paper and we would like to address your concerns below.
>
> > The use of adversarial training and data augmentation for defense is not entirely new
>
> We highlight that while adversarial training
> and data augmentation are not new,
> the proposal of adversarial augmentation
> is novel and effective,
> as it forms an effective defense
> against unlearnable example attacks.
> It also breaks the perception that data augmentations
> were not effective countermeasure against unlearnable examples.
> Finally,
> it outperforms the state-of-the-art methods
> against existing unlearnable example attacks.
>
> > The paper could benefit from exploring additional evaluation of UEraser and UEraserLite, particularly in terms of computational efficiency/time cost compared with existing defenses
>
> Thank you for your constructive feedback.
> We have added the time cost of proposed method
> versus other defense methods in the Appendix F.
>
> > a more comprehensive theoretical analysis of why UEraser works and under what conditions it is most effective could strengthen the paper
>
> We thank the reviewer for the suggestion.
> The main contribution of this paper
> is the proposal of adversarial augmentation,
> empirically show its ability
> to mitigate the impact of unlearnable example attacks,
> and shed light to the inadequacy of the existing attacks.
> We hope to turn our focus to its theoretical analysis
> in the future.

---

### Official Review · Reviewer_iWnX · 2023-10-31

**Soundness:** 2 fair
**Presentation:** 2 fair
**Contribution:** 2 fair
**Rating:** 1
**Confidence:** 5

**Summary:**

Unlearnable example attacks refer to data poisoning techniques used to safeguard public data against unauthorized training of deep learning models. These methods introduce subtle perturbations to the original images, making it challenging for deep learning models to effectively learn from such training data. Current research indicates that adversarial training can partially mitigate the impact of unlearnable example attacks, while common data augmentation methods are ineffective against these poisons. However, adversarial training is computationally demanding and can lead to significant accuracy loss.

This paper presents the UEraser method, which surpasses existing defenses against various state-of-the-art unlearnable example attacks. UEraser achieves this through a combination of effective data augmentation techniques and loss-maximizing adversarial augmentations. Unlike current state-of-the-art adversarial training methods, UEraser employs adversarial augmentations that extend beyond the assumed 'p perturbation budget' used by current unlearning attacks and defenses. This approach enhances the model's generalization ability, thereby safeguarding against accuracy loss. UEraser effectively eliminates the unlearning impact through loss-maximizing data augmentations, restoring trained model accuracies.  On challenging unlearnable datasets like CIFAR-10, CIFAR-100, SVHN, and ImageNet-subset, generated using various attacks, UEraser achieves good results.

**Strengths:**

1. This paper highlights that unlearnable examples can be mitigated through various data augmentation techniques, potentially leading to the generation of more resilient unlearnable examples when facing adaptive poisoning.

**Weaknesses:**

1. After carefully conducting experiments of the UEraser using the official code provided by the authors, a notable disparity emerged in comparison to the reported results in the paper. On the CIFAR-10 datasets, specifically, for EM poisons, the best accuracy achieved during training was 74.46\%, while the accuracy in the final epoch was 68.46% (25\% lower than reported). In the case of LSP poisons, the best training accuracy reached 90.23\%, with a final epoch accuracy of 73.43\% (12\% lower than reported). This observation indicates that although the UEraser appeared effective during mid-training, the models eventually converged to shortcuts present in the unlearnable examples. Given that there is often no clean validation dataset available, many papers on unlearnable examples (UE) typically report the accuracy achieved in the last training epoch. Furthermore, with a fixed learning rate of 0.01 throughout the training process, I've observed that the accuracy of models trained on clean images struggles to converge to a satisfactory level, reaching approximately 92% on CIFAR-10 (2% lower than reported) and around 70% on CIFAR-100 (4% lower than reported) in my experiments. Therefore, it is recommended that the authors thoroughly review their results and consider reporting the accuracy in the last epoch for a more equitable comparison.

2. In the experiments conducted with the UEraser-Max approach, the performance on the EM and LSP attacks shows a gap compared to the reported results on the CIFAR-10 dataset (EM: 62.25\%, 33\% lower than reported; LSP: 88.33\%, 6\% lower than reported). Interestingly, the use of a fixed learning rate of 0.01 (too large to converge to the optimal model when conducting UEraser-Max) is not likely to reach an accuracy of 95.24% for EM poisons, which is even 0.5% higher than what is typically achieved in standard training on **clean** CIFAR-10.

2. It's essential to consider the standard evaluation settings and practices in the field. The results on ImageNet-subset are not convincing, as most operations on ImageNet-subset in papers related with Unlearnable Examples do not resize the images to $32 \times 32$. Instead, they follows the default data augmentations, and images are resized to $224 \times 224$ during training, it would be advisable to align with these practices in your experiments for better comparability with previous works. For your reference, usually, the clean performance on the ImageNet-subset should be around 80\%, and the performance of ISS facing several UE methods is around 55\%.

3. Training on CIFAR-10, CIFAR-100, and SVHN datasets takes a similar amount of time. To provide a more comprehensive evaluation of the proposed method's performance and robustness, it would be valuable to expand the experimental analysis to include CIFAR-100 and SVHN, similar to the approach taken for CIFAR-10. This extended evaluation would help assess how well the model generalizes across different datasets and under various UE attack methods.

4. The proposed methods do not exhibit robustness to adaptive poisoning. Table 4 demonstrates that when faced with adaptive poisoning (UEraser-Max), the defensive performance experiences a significant drop, ranging from 15% to 30%. To facilitate a fair comparison, it would be beneficial to report the performance of adaptive poisoning when facing ISS.

5. The concept presented in this work is not particularly innovative, and the achieved performance heavily relies on empirical augmentations for defense, which can be significantly undermined by adaptive poisoning.

**Questions:**

See weaknesses above.

---

> ### Author Response · Authors · 2023-11-15
> **Rebuttal by Authors**
>
> Thank you for reviewing our paper and we would like to address your concerns below.
>
> > with a fixed learning rate of 0.01 throughout the training process, I've observed that the accuracy of models trained on clean images struggles to converge to a satisfactory level
>
> > the use of a fixed learning rate of 0.01 (too large to converge to the optimal model when conducting UEraser-Max) is not likely to reach an accuracy of 95.24% for EM poisons
>
> > a notable disparity emerged in comparison
> > to the reported results in the paper ...
>
> We sincerely apologize for the oversight in the paper,
> where instead of a fixed 0.01 learning rate,
> we used the cosine annealing schedule
> with an initial learning rate of 0.01
> in experiments by default.
> The paper has been updated to reflect this,
> and we appreciate your diligence
> in identifying this discrepancy,
> and we are committed
> to providing accurate and reliable results.
>
> We appreciate your keen interest
> in our paper
> and the effort you've invested
> in reproducing the results.
> We would like to engage with you
> in the discussion period
> to address the questions raised
> and help you reproduce the results.
> If permissible by the conference guidelines,
> we could set up an anonymized Google Colab
> to ensure its reproducibility.
>
> > Are results on ImageNet-subset resized to 32x32?
>
> Thank you for pointing this out to us.
> Section 4 incorrectly states
> that the images are resized to 32x32,
> instead they are resized to 224x224,
> as originally stated in Table 8 of Appendix A.1.
> The accuracy differences from previous papers
> can be attributed
> to the smaller training set size (200 images per class).
>
> > ... extended evaluation would help assess how well the model generalizes across different datasets and under various UE attack methods.
>
> We thank the reviewer for the suggestion,
> and given the time available during discussion period
> and limited computational resources,
> we will expand the experimental analysis
> on CIFAR-100 in an updated version.
>
> > The proposed methods do not exhibit robustness to adaptive poisoning using UEraser-Max.  To facilitate a fair comparison, it would be beneficial to report the performance of adaptive poisoning when facing ISS.
>
> We have added adaptive poisoning and defenses with ISS
> in Appendix E.
> Additionally,
> achieving over 80% accuracy under adaptive poisoning of our method
> should not be perceived negatively,
> which is in line with adaptive attacks
> using adversarial training and ISS.
>
> > The concept presented in this work is not particularly innovative, and the achieved performance heavily relies on empirical augmentations for defense, which can be significantly undermined by adaptive poisoning.
>
> We highlight that the novelty of UEraser variants
> is the proposal of adversarial augmentation,
> and its ability to mitigate the impact of unlearnable examples.
> We also point out that the key takeaway of our paper
> is that all existing $\ell_p$ bounded attacks
> cannot effectively protect the data against learning,
> which is a significant finding
> that needs urgent attention from the community.
>
> We hope to engage with the reviewer further
> for us to address your concerns more thoroughly.

---

> ### Comment · Reviewer_iWnX · 2023-11-15
> **Updated Comments**
>
> 1. The response is not satisfactory to me. I've experimented with various combinations of optimizer setups and schedules, but none of the results have matched the reported performance. Moreover, if the fundamental setup is unclear, it raises concerns about the credibility of the reported results. Moreover, the choice of an initial learning rate of 0.01 is questionable when experiments are conducted using a cosine annealing schedule. In general, other learning problems on CIFAR-10, such as adversarial training, commonly utilize an initial learning rate of 0.1 when employing the SGD optimizer. It's worth noting that this paper doesn't make any claims about unusual setups in this regard.
>
> 2. Another concerning point is in the comparison of training on CIFAR-10 with EM in Table 2 and Table 6. I don't think that changing the backbone from ResNet-18 to ResNet-50 would result in a drop of about 5%.
>
> 3. The additional Adversarial Training results lack convincing evidence for both CIFAR-100 and ImageNet-subset. In my implementations, for ImageNet-subset (200 images per class), performance reaches 44% for both EM and REM, and the proposed method is comparable to adversarial training in terms of computation. On the CIFAR-100 dataset, performance exceeds 55%, higher than reported ones.
>
> 4. Moreover, the outcomes on ImageNet-subset, particularly when training on clean data, appear not practical. If you consider 200 images per class, I am certain that the accuracy cannot attain 72%. ISS [1] reported similar results, with standard training on clean ImageNet-subset yielding 62%.
>
> 5. The claimed effectiveness of the proposed methods seems overstated due to 1) Limited experiments on datasets other than CIFAR-10; 2) Subpar performance on ImageNet-subset, potentially tailored for datasets with small resolutions.
>
> 6. The reported results appear to sidestep more robust defense methods, such as median filtering, which can achieve 85% on the OPS, surpassing the proposed method.
>
> 7. When confronted with adaptive poisoning, the performance is lower than adversarial training (80% vs. 85%).
>
> 8. Using augmentation as a defensive method lacks compelling support. This type of augmentation-based approach has been extensively discussed in related attacks. The novelty of this paper seems to fall below the acceptance standards of ICLR.
>
> In summary, I highly recommend that the authors conduct a thorough examination of the experimental setups, provide detailed implementations, and conduct more comprehensive experiments.
>
> [1] Zhuoran Liu, Zhengyu Zhao, and Martha Larson. **Image shortcut squeezing: Countering perturbative availability poisons with compression.** In *Proc. Int’l Conf. Machine Learning*, 2023

---

> > ### Author Response · Authors · 2023-11-16
> > **Response to the reviewer's additional concerns and current unsubstantiated claims**
> >
> > 1. While the choice of the default training hyperparameters may differ from existing work, yet the choices are subjective and not pertinent to the discovery made in this paper. We can certainly help in regard of reproducibility.
> > 2. The results are correct and we do not believe this invalidates our claims, as the training runs may exhibit high variances in the peak accuracies.
> > 3. The differences again may be attributed to the choice of specific hyperparameters. We kindly request your code to run adversarial training and we would gladly reproduce the attacks on your implementation to allay your concerns.
> > 4. Thank you for raising this to us. The result we reported for the clean baseline used more images, and 200 images per class should indeed not reach 72% on clean training, we apologize for our negligence in this data and are rerunning the experiment.
> > 5. As mentioned in our earlier reply, (1) we are extending the experiments on CIFAR-100; (2) we kindly note that our ImageNet subset used 224x224 images and the accuracy reaches 45.67% on UEraser, which exceeds your run on EM (44%), despite reviewer's belief that our hyperparameters could be suboptimal.
> > 6. We kindly request the reviewer to make the claim of 85% on OPS with median filtering available to us. A reproducible implementation would be much appreciated.
> > 7. The reviewer failed to outline why may want to compare the adaptive performances of adversarial training and UEraser variants, where the game could be a prisoner's dilemma, with the defender choosing UEraser would win most of the time. It is also clear that the adaptive performances of adversarial training (specifically, REM attack), is largely inferior to UEraser.
> > 8. We point out that the key novelty of this paper lies in the proposal of adversarial augmentations, rather than specific designs of the augmentation policies.